# Identification of Alternative Mitochondrial Electron Transport Pathway Components in Chickpea Indicates a Differential Response to Salinity Stress between Cultivars

**DOI:** 10.3390/ijms21113844

**Published:** 2020-05-28

**Authors:** Crystal Sweetman, Troy K. Miller, Nicholas J. Booth, Yuri Shavrukov, Colin L.D. Jenkins, Kathleen L. Soole, David A. Day

**Affiliations:** College of Science & Engineering, Flinders University, GPO Box 5100, Adelaide SA 5001, Australia; troy.miller@flinders.edu.au (T.K.M.); nick.booth@flinders.edu.au (N.J.B.); yuri.shavrukov@flinders.edu.au (Y.S.); colin.jenkins@flinders.edu.au (C.L.D.J.); kathleen.soole@flinders.edu.au (K.L.S.); david.day@flinders.edu.au (D.A.D.)

**Keywords:** chickpea, salinity, alternative oxidase, alternative NAD(P)H dehydrogenase, gene expression, co-expression, mitochondria, enzyme activity, high Na accumulator

## Abstract

All plants contain an alternative electron transport pathway (AP) in their mitochondria, consisting of the alternative oxidase (AOX) and type 2 NAD(P)H dehydrogenase (ND) families, that are thought to play a role in controlling oxidative stress responses at the cellular level. These alternative electron transport components have been extensively studied in plants like Arabidopsis and stress inducible isoforms identified, but we know very little about them in the important crop plant chickpea. Here we identify AP components in chickpea (*Cicer arietinum*) and explore their response to stress at the transcript level. Based on sequence similarity with the functionally characterized proteins of *Arabidopsis thaliana*, five putative internal (matrix)-facing NAD(P)H dehydrogenases (*CaNDA1-4* and *CaNDC1*) and four putative external (inter-membrane space)-facing NAD(P)H dehydrogenases (*CaNDB1-4*) were identified in chickpea. The corresponding activities were demonstrated for the first time in purified mitochondria of chickpea leaves and roots. Oxidation of matrix NADH generated from malate or glycine in the presence of the Complex I inhibitor rotenone was high compared to other plant species, as was oxidation of exogenous NAD(P)H. In leaf mitochondria, external NADH oxidation was stimulated by exogenous calcium and external NADPH oxidation was essentially calcium dependent. However, in roots these activities were low and largely calcium independent. A salinity experiment with six chickpea cultivars was used to identify salt-responsive alternative oxidase and NAD(P)H dehydrogenase gene transcripts in leaves from a three-point time series. An analysis of the Na:K ratio and Na content separated these cultivars into high and low Na accumulators. In the high Na accumulators, there was a significant up-regulation of *CaAOX1*, *CaNDB2*, *CaNDB4*, *CaNDA3* and *CaNDC1* in leaf tissue under long term stress, suggesting the formation of a stress-modified form of the mitochondrial electron transport chain (mETC) in leaves of these cultivars. In particular, stress-induced expression of the *CaNDB2* gene showed a striking positive correlation with that of *CaAOX1* across all genotypes and time points. The coordinated salinity-induced up-regulation of *CaAOX1* and *CaNDB2* suggests that the mitochondrial alternative pathway of respiration is an important facet of the stress response in chickpea, in high Na accumulators in particular, despite high capacities for both of these activities in leaf mitochondria of non-stressed chickpeas.

## 1. Introduction

Mitochondria in all higher aerobic organisms are essential for the energy status of the cell and also play key roles in oxidative stress generation and management. Plant mitochondria have a branched electron transport chain (mETC): in addition to the classical mETC consisting of large complexes that transfer electrons from intramitochondrial NADH to oxygen, coupled to proton translocation and ATP synthesis, plant mitochondria contain an alternative pathway (AP) that does not conserve energy [1,2]. The AP consists of several single-subunit type II NAD(P)H dehydrogenases (NDs) on both the outside (NDBs) and inside (NDAs and NDCs) of the inner mitochondrial membrane, and an alternative oxidase (AOX) that accepts electrons from the ubiquinol pool and competes with the cytochrome pathway [3,4,5,6,7,8]. While there has been much speculation on the role of the AP, a consensus has emerged that it plays a key role in moderating the generation of reactive oxygen species (ROS) in the mETC and affects oxidative stress signaling in plant cells through linkages with key transcription factors [9,10,11,12]. Additionally, the AP may modulate other metabolic pathways and growth, through the regulation of cellular energy status and respiratory metabolite pools [7,13,14,15,16]. Overexpression of AP components can confer enhanced tolerance of environmental stresses in plants [17,18,19,20,21,22].

The genomic organization of AP genes, especially AOX, varies between species [23,24]. In many dicot species, exemplified by the model plant *Arabidopsis thaliana*, AOX is encoded in a small gene family consisting of several AOX1 genes and maybe one AOX2 gene, with one of the AOX1 genes the main expressed isoform, but only in the presence of stress. Monocot species, including important crop plants such as rice and wheat, lack functional AOX2 isoforms. The legume family appears to be another exception, with all species examined to date having multiple forms of AOX2, with at least one constitutively expressed and another preferentially transcribed to a high level in shoots [25]. In Arabidopsis, the NDs are also encoded in a small gene family, with two NDA genes, four NDB genes, and an NDC gene [4,26]. NDC1 and some of the NDA and NDB proteins are also found in other organelles [27]. The genomic structure of the ND genes in other species is less understood, although barley and rice ND genes have been identified and partially characterized [28].

Here we have investigated the AP in the important crop legume, chickpea, including the identification of ND genes and determination of those genes that are stress responsive. There was a marked difference in the expression of specific AP genes depending on whether cultivars accumulated high or low levels of Na in their leaves, suggesting that this pathway is important in tissues that accumulate Na. This information is important for our understanding of the role of mitochondria in tissue-specific functions and environmental stress responses in legumes.

## 2. Results

### 2.1. Type II NAD(P)H Dehydrogenase Activities in Chickpea

Activities for internal and external NAD(P)H dehydrogenases were measured in mitochondria purified from chickpea leaves and roots (Figure 1). In leaf mitochondria, the rate of external NADH oxidation was greater than 200 nmol O_2_ min^−1^ mg^−1^ protein and more than half of this rate was calcium-independent. External NADPH oxidation was approximately 100 nmol O_2_ min^−1^ mg^−1^ protein and completely calcium-dependent. In contrast, external NADH and NADPH oxidation rates in mitochondria from roots were lower by a factor of almost 10 and less dependent on calcium. Internal (matrix-generated) NADH oxidation rates, measured as rotenone-insensitive malate or glycine oxidation, were lower than those for external NADH or NADPH in leaf mitochondria. This was not likely due to a limitation in the rate of NADH generation in the matrix, as higher rates of malate and glycine oxidation were observed in the absence of rotenone (i.e., when Complex I was also active) and may reflect the poorer affinity of this pathway for NADH as observed in other species [29]. However, both rotenone-sensitive and rotenone-insensitive malate and glycine oxidation rates were much lower in root mitochondria.

### 2.2. Type II NAD(P)H Dehydrogenase Genes in Chickpea

#### 2.2.1. Gene Identification

To identify candidate genes encoding ND proteins, *C. arietinum* sequences were extracted from the non-redundant protein sequence database (NCBI), using ND protein sequences previously characterized from *A. thaliana* [26]. Nine putative ND orthologs mapped to unique regions of the *C. arietinum* genome (Table 1) and these were classified according to *A. thaliana* nomenclature where possible: four NDA types *CaNDA1-4* (i.e., internal-facing), four NDB types *CaNDB1-4* (i.e., external-facing) and one NDC type *CaNDC1*. Sub-cellular localization was predicted based on the presence of N-terminal signal peptides (Appendix A). Despite some disagreement between results from the different targeting tools, most NDA and NDB proteins are likely to target to mitochondria while NDC1 is probably dual targeted to the mitochondria and chloroplasts, as also seen in Arabidopsis [27]. Discrepancies between programs also highlights the need for experimental determination of cellular location of these proteins. Notably, some programs did not predict mitochondrial targeting for some Arabidopsis NDs despite experimental evidence for this [27,30].

Each of the CaND amino acid sequences contained the conserved FAD-binding domain (Appendix A) and all CaNDB sequences contained an EF-hand domain, which is characteristic of NDB proteins from other plants and the site of calcium binding [31,32,33]. Clustal alignments of the highly conserved NAD(P)H binding domains revealed no differences between the expected substrate preferences for chickpea NDs compared to Arabidopsis NDs, whereby all NDAs and NDBs (except NDB1) contained a glutamate residue that is characteristic of NADH preference at its binding site [34]. Additionally, like Arabidopsis, the chickpea NDB1 contained a glutamine at this site and is likely to preferentially bind NADPH (Appendix A).

Based on amino acid sequence identity (Table 1), it was not possible to assign any of the four CaNDAs as specific orthologues to AtNDA1 or AtNDA2. CaNDA1 and CaNDA2 shared 72–74% identity to both Arabidopsis NDAs, and CaNDA3 and CaNDA4 were approximately 66% identical to both Arabidopsis NDAs. Similarly, it was not possible to distinguish whether CaNDB2 or CaNDB3 were more like AtNDB2 or AtNDB3, since both chickpea proteins showed approximately 70% identity to both Arabidopsis counterparts. CaNDB4 shared 66–68% identity to AtNDB2, AtNDB3 and AtNDB4. However, CaNDB1 clearly showed the highest identity with AtNDB1, and CaNDC1 also showed the highest identity with AtNDC1.

CaND genes were found on five of the eight chickpea chromosomes, although the majority were located on chromosome 6 (*CaNDA3*, *CaNDB1*, *CaNDB3*, *CaNDB4* and *CaNDC1*). *CaNDB1* and *CaNDB4* were only 11 kb apart, towards the 3′ end of chromosome Ca6. *CaNDA3* and *CaNDB3* were also close to each other (~250 kb) but considerably upstream of *CaNDB1* and *CaNDB4*. As shown previously, the three *CaAOX2* genes co-localized on chromosome Ca6 [25], but were approximately 1.1 Mb upstream of *CaNDB3*. All other CaND genes, *CaNDA1, CaNDA2*, *CaNDA4* and *CaNDB2* as well as the *CaAOX1* gene [25], were found on separate chromosomes (Table 1; Ca5, 2, 4, 1 and 8, respectively).

Chickpea ND genes had similar exon structures to Arabidopsis (Appendix A). *CaNDA1* and *2* had similar structures to the *AtNDA* genes, with eight introns, but *CaNDA3* and *CaNDA4* each had an extra intron at the start that contained only untranslated regions. A splice variant of the *CaNDA2* gene was also predicted, whereby exons 5 and 6 were lost. This variant was termed *CaNDA5*. *CaNDB1* had 10 exons compared to the nine exons of *AtNDB1* due to an additional intron within exon 5. *CaNDB2*, *3* and *4* each had 10 exons as per *AtNDB2* and *3* and none of the *CaNDB* genes had the six-exon structure of *AtNDB4*. *CaNDC1* had 11 exons compared to the 10 exons of *AtNDC1*, due to an additional intron within the final exon. *CaND* genes typically had extended introns compared to Arabidopsis, some as large as 1-2.5 kb (found in *CaNDA2*, *CaNDB1*, *CaNDB2*, *CaNDB3* and *CaNDC1*). The fourth introns of *CaNDA2* and *CaNDA5* were considerably longer, at approximately 5.5 and 6 kb, respectively.

#### 2.2.2. Differential Expression of AP Components in Chickpea Tissues

Tissue-specific expression patterns were explored (Table 2 and Appendix A) using publicly available RNAseq data from an experiment with 15-day old seedlings [35] and from another experiment with plants at various developmental stages [36]. These were subsequently confirmed by qRT-PCR on our samples collected during mitochondrial isolation experiments. AOX gene transcript levels were also analysed by qRT-PCR and tissue-differential patterns of expression matched to those seen previously [25].

From these datasets, *CaNDA1* was the most highly expressed *CaNDA* gene in the shoot/leaf but was missing entirely from roots (Table 2, Figure 2). This is similar to the *AtNDA1* gene in Arabidopsis [30]. The *CaNDA2* gene was expressed in all tissues but generally higher (or similar) in the root compared to leaf, also similar to the Arabidopsis *AtNDA2* gene [30]. *CaNDA2* had a potential alternative splice variant, so primers were designed to target both variants individually, with the latter designated *CaNDA5*. Transcripts could be detected for both of these variants, although *CaNDA2* levels were approximately 10-fold those for *CaNDA5*. No RNAseq contigs for the *CaNDA3* could be found, but this transcript was detected in low levels using qRT-PCR and was significantly lower in roots compared to leaves. In the Chickpea Transcriptome DataBase (CTDB), the *CaNDA4* gene was represented by three small contigs rather than one complete contig, none of which were detected in the tissue libraries, although transcript levels were detected in most tissues from Kudapa et al. [36], where they were lower in roots than in leaves. This was confirmed using qRT-PCR, with the gene expressed at a low level similar to that of *CaNDA3* (Figure 2).

The four NDB genes also showed transcriptional variation. *CaNDB3* was the most highly expressed *CaNDB* gene in all tissues, while *CaNDB1* was less abundant, especially in roots (Table 2, Figure 2). In contrast, the *AtNDB1* gene was reported to be equally expressed in leaf and root tissues of Arabidopsis [30]. *CaNDB2* and *CaNDB4* were not detected in any tissue from RNAseq datasets but were present at low levels in qRT-PCR data, where expression was significantly lower in roots. *CaNDC1* was expressed in all tissues but lower in (or absent from) roots, unlike the *AtNDC1* gene, which was expressed equally in leaves and roots of Arabidopsis.

#### 2.2.3. Expression of AP Genes in Response to Salinity Stress Varies between Cultivars and Is Related to Na Accumulation

To document the stress-responsiveness of ND genes in chickpea, mRNA transcripts of leaf samples from six chickpea cultivars exposed to salinity stress (90 mM) were measured. The growth response of these cultivars to salinity has been reported previously [37] and for convenience is summarized here in Appendix A. Cultivars were classified as salt tolerant, sensitive or moderately sensitive based on the impact on growth and visible symptoms [37]. Cultivars were also classed as high or low salt accumulators based on Na:K ratios.

Leaves were harvested on days 5, 9 and 15 (T5, T9 and T15) after the final NaCl application, and expression levels of the *CaAOX* and *CaND* genes were quantified using qRT-PCR (Figure 3 and Appendix A). As seen previously [25], *CaAOX2a* was the most highly expressed alternative oxidase gene in chickpea leaves, whilst *CaAOX1* was the most stress responsive. Of the NDs, *CaNDA1* was the most highly expressed followed by *CaNDB3* and *CaNDC1*, but none of these were consistently upregulated by stress. Only *CaNDB2* showed a high level of induction across a number of time points, with 3- to 10-fold increased expression in Hattrick, ICC12726, Slasher and Yubileiny after 9 or 15 days (or both) of salt exposure. This gene was otherwise expressed at very low levels and undetected in RNAseq experiments (see above). Some other *CaND* genes showed transcript up-regulation in specific cultivars and at specific time points including *CaNDA1*, which was up-regulated 3- to 4-fold in Hattrick and Slasher after 15 days; *CaNDA2*, which was up-regulated two-fold in ICC12726 and Yubileiny after 15 days; *CaNDA3*, which was temporarily up-regulated seven-fold in Slasher after 9 days; *CaNDA4*, up-regulated four- and 12-fold in ICC12726 and Yubileiny after 15 days, respectively; *CaNDB1*, up-regulated two-fold in ICC12726 at 15 days of salt-exposure; *CaNDB3*, up-regulated two-fold in Slasher and Yubileiny after 15 days; and *CaNDB4*, up-regulated in several cultivars but not significantly, except for a three-fold increase in ICC12726 after 15 days and four-fold increase in Rupali after 9 days. From this it can be seen that there is genotype- and condition-specific regulation of the different *CaND* genes, with *CaNDB2* the most commonly up-regulated.

An analysis of Na content in the cultivars showed that they could be grouped into high and low Na accumulators (Appendix A). The transcriptional response of AP genes at different time points was then re-analysed (Figure 4). At 15 days *CaAOX1*, *CaNDB2*, *CaNDB4* and *CaNDC1* were more highly upregulated in the higher Na accumulators compared to the lower accumulators. Despite the correlation with leaf Na:K, there was no significant correlation of AP gene expression response with plant biomass [37] after one month of salt (not shown).

### 2.3. Type II NAD(P)H Dehydrogenase Genes of Other Legumes

To determine whether other legumes demonstrate similar ND expression patterns to chickpea, orthologs from 15 additional legume species were identified (Appendix A). In general, there were similar numbers of ND gene family members across legume species, typically between 7 and 9 genes comprised, on average, of three *NDA* genes, four *NDB* genes and one *NDC* gene. Notable exceptions were *Glycine max*, which contained 14 putative ND genes including six *NDA*s, seven *NDB*s and one *NDC* and *Prosopis alba*, the white carob tree, which contained 10 putative ND genes including four *NDA*s, four *NDB*s and two *NDC*s. *G. max* is a partially diploidized tetraploid, which may explain the larger set of ND genes, although this species does not have an expanded AOX gene family compared to other legumes [25]. Many of the ND proteins are predicted to be targeted to mitochondria (Appendix A) although this is yet to be experimentally determined.

According to publicly available transcriptomic data from *G. max*, *Medicago truncatula*, *Lotus japonicus*, *Vigna unguiculata*, *Cajanus cajan* and *Phaseolus vulgaris*, *NDA1* and *NDC1* transcripts are also largely absent from root tissues, while *NDB1* is less expressed in roots compared to leaves or shoots (Appendix A).There are limited transcriptomic datasets available for abiotic stress experiments in legumes, and of these, many lack information for the entire set of ND genes (Appendix A). Data from Genevestigator (Hruz et al., 2008) suggest that *MtNDA3* is a stress-responsive gene in *M. truncatula*, becoming up-regulated in nine stress-imposed conditions, including salt, drought, cold, ozone and low phosphorous treatments, while *MtNDA2* was up-regulated by drought and *MtNDB3* by ozone [38,39,40,41]. Data from Lotus Base suggested that *LjNDA3* is also up-regulated by drought in *L. japonicus*, and in this case there were some data available for *LjNDB2*, which is up-regulated by drought and salt treatments [42,43,44].

### 2.4. Co-Expression of AP Genes

The pattern of *CaNDB2* transcript up-regulation was similar to that of *CaAOX1*, suggesting that there may be a degree of stress-induced co-regulation of these two genes in chickpea, as seen for *AtAOX1a* and *AtNDB2* in Arabidopsis [45]. To explore further the possibility of co-regulation of the *CaAOX1* and *CaNDB2* genes, a correlation analysis with linear regression was completed, using transcript data from the salt experiment across all cultivars and treatments, at individual time points (Figure 5 and Appendix A). Positive correlations were found when using either relative transcript level data or fold change data, particularly at the two later timepoints T9 and T15 (Figure 5B and Figure 4F). Other AP genes that showed positive linear correlations with respect to their salinity response included *CaAOX2A*, *CaAOX2B*, *CaNDA1* and *CaNDC1* (Appendix A) and these also shared tissue-specific expression patterns, i.e., more highly expressed in shoots than roots (Figure 2, Table 2). *CaNDA2*, *CaNDA3*, *CaNDA4* and *CaNDB1* also showed positive correlations during the response to salt.

To further explore the potential co-regulation of *CaAOX1* and *CaNDB2*, 2 kb of sequence upstream of the mRNA transcriptional start sites were scanned for the presence of elements highlighted by Thirkettle-Watts et al. [46] in *G. max* AOX promoter regions and by Clifton et al. [45] in *A. thaliana AtAOX1A* and *AtNDB2* promoter regions, as well as for the presence of putative cis-acting regulatory elements in the PlantCARE database [47] (Appendix A). The upstream region of *CaAOX2A* was also included in these searches as a negative control, however only 1.6 kb could be used due to the presence of a neighbouring gene. *CaAOX1*, *CaNDB2* and *CaAOX2A* all shared many regulatory elements from the PlantCARE database, therefore making it difficult to identify particular elements that may be responsible for salt-induced co-regulation of *CaAOX1* and *CaNDB2*. However, elements present in AP genes of *A. thaliana* and *G. max* [45,46] were more common in the two stress-inducible genes *CaAOX1* (13 elements) and *CaNDB2* (6 elements) compared to the highly expressed *CaAOX2A* gene (2 elements), indicating that this warrants further experimental analysis.

To determine whether other legumes share similar co-expression patterns of AP genes during salinity treatments, publicly available transcriptomic data were revisited. Unfortunately, transcript level information for NDB2 and some other ND genes are missing from many legume datasets, due to the absence of probe sets on microarray chips and possibly below limits of detection for sequence reads. However, there are some data available for *L. japonicus* and a strong positive correlation was observed between *LjAOX1* and *LjNDB2* expression in salinity experiments (adjusted R^2^ = 0.86, *p* = 0.000; Appendix A).

### 2.5. Maintaining Photosynthetic Rates during Salinity Stress

Chickpea cultivars that did not show enhanced *CaNDB2* expression in response to salt, namely Genesis836, ICC12726 and Rupali, also showed the biggest inhibition of the rate of photosynthesis (Figure 6). Based on correlation analyses across all cultivars (Appendix A), there was a positive relationship between *CaNDB2* fold change at 9 days salinity treatment and photosynthesis at 4 weeks (R^2^ = 0.77, *p* = 0.072). A very similar relationship was also seen for *CaNDB3* at 15 days. There was an even stronger positive correlation between *CaAOX1* fold change at both 5 days and 9 days salinity treatment and photosynthesis at 4 weeks (R^2^ = 0.99, *p* < 0.001 and R^2^ = 0.89, *p* = 0.02, respectively). *CaNDC1* fold change at 15 days also showed a weak positive correlation with photosynthesis, although not significant (R^2^ = 0.71, *p* = 0.111).

## 3. Discussion

The aim of this study was to identify the genes encoding AP components in chickpea mitochondria and to determine their expression in different tissues and in response to an environmental stress. We have identified, for the first time, the genes encoding members of the type2 alternative NAD(P)H dehydrogenases of the alternative mitochondrial electron transport chain in chickpea, and report on the activities of the corresponding proteins in mitochondria from leaves and roots. We have also examined the effect of salt stress on AP gene expression in leaves and found that specific gene responses are dependent on tissue Na content.

### 3.1. NAD(P)H Oxidation by Isolated Mitochondria

Rates of external NADH oxidation in chickpea leaf mitochondria were substantially higher (2–4 times) than those reported for *A. thaliana* leaf mitochondria, which under control conditions have very little AOX activity [13,20,22,48]. Rates were similar to those of soybean, tobacco and potato leaf mitochondria, all which have substantial AOX activity [5,49,50]. Similarly high rates of external NADH oxidation were seen in mitochondria from Arabidopsis transgenic lines over-expressing AOX and NDB2, which also have much higher AOX activity [22]. This might indicate that high AOX capacity enables high external NADH oxidation rates in leaf mitochondria, although this is not the case in non-photosynthetic tissues, since mitochondria from potato tubers have no AOX activity but readily oxidize external NADH [51].

Mitochondria from roots showed much less capacity than those from leaves for both external NAD(P)H and internal NADH oxidation, with the latter measured as oxidation of NAD-linked substrates. The lower rate of glycine oxidation in root mitochondria was expected, since glycine decarboxylase (GDC) genes are light dependent [52], but the much lower rates of malate oxidation in both the presence and absence of rotenone in roots are unusual in general [51]. Mitochondria from roots of soybean can oxidize both external NADH and malate rapidly and at rates similar to those in cotyledons [53]. The lower NAD(P)H oxidation rates in roots of chickpea were correlated with lower expression of all NDB genes and most of the NDA genes, as well as NDC, relative to leaves. Succinate oxidation rates were also much higher in leaves than in roots, as was AOX capacity [25]. Together, the results point to a far greater respiratory capacity in the leaves of chickpea, compared to roots.

### 3.2. Expression of AP Genes in Different Tissues

Our qRT-PCR analysis largely confirmed patterns of *CaND* expression as found in RNAseq databases and patterns of *CaAOX* expression reported in a previous study [25]. Overall, many of the AP genes were more highly expressed in leaves compared to roots by at least two-fold, the exceptions being *CaAOX2D*, *CaNDA2* and *CaNDA5*, which were lower in roots but not significantly so. *CaAOX2A* and *CaNDA1* in particular, but also *CaAOX2B*, *CaNDB1* and *CaNDC1*, were highly expressed in leaves but essentially absent in roots. *CaNDA3*, *CaNDA4*, *CaNDB2* and *CaNDB4* were all expressed at very low levels in both organs, explaining the lack of data for these genes in RNAseq datasets; for each of these genes, transcript levels were significantly lower in roots than leaves. Public RNAseq data show that low expression of *NDA1* and *NDC1* in the roots seems to be a consistent feature of legumes. *NDA1*, *NDC1* and *AOX2A* were also co-expressed across the six cultivars and three time points in the salinity stress experiment, perhaps pointing to cooperation between the three enzymes in photosynthetic tissues.

Why chickpea and other legumes (especially soybean) have such a large cohort of *ND* genes, compared to Arabidopsis, is not obvious, but it makes nomenclature rather difficult. The nomenclature used here is based partly on homology of the CaND protein sequences with Arabidopsis counterparts and their patterns of gene expression. CaNDB1 and CaNDC1 aligned well with the specific isoforms of Arabidopsis ND proteins based on sequence identity and tissue gene expression patterns. CaNDA1-4 did not, although CaNDA1 and 2 were more similar to the *A. thaliana* AtNDA protein sequences than CaNDA3 and 4. CaNDB2, 3 and 4 could not be associated with a particular *A. thaliana* AtNDB protein based on sequence identity nor tissue-specific gene expression patterns. CaNDB2 and CaNDB3 aligned with similar identity to both AtNDB2 and AtNDB3, and to counterparts in rice and barley [28]. Like *AtNDB2*, *CaNDB3* was the most highly expressed isoform at the transcript level. However, *CaNDB3* was not as stress-responsive; in that respect *CaNDB2* was more similar to *AtNDB2*.

### 3.3. Stress-Responsive AP Genes in Chickpea

Some AP components in chickpea were stress-responsive (especially *CaAOX1* and *CaNDB2*), but their up-regulation was time-dependent and temporary, as seen for some other species [28]. Analysis of elemental data from [37] showed that the six cultivars of chickpea could be separated into high and low accumulators of Na into leaf tissue, although lower leaf Na did not equate to improved tolerance, an observation made previously in chickpea [54,55]. There was a significant up-regulation of *CaAOX1, CaNDB2, CaNDB4, CaNDA3* and *CaNDC1* in the high Na accumulators which was absent in the low accumulators. This suggests that a stress response occurred in tissues with high Na levels, resulting in the upregulation of genes whose protein products encode a non-phosphorylating mETC. The signals mediating this upregulation remain to be identified, but it is possible that increased ROS or demand for osmolytes to help osmotic adjustment may have led to enhanced AP capacity in cells accumulating high Na. Such a response has been observed in Arabidopsis under salinity stress, which also accumulates Na at high levels [20]. Regardless of the mechanism, in the current study the upregulation of these genes in the high Na accumulators did not correlate with improved biomass during salinity stress in these cultivars.

Coordinated expression of *CaAOX1* and *CaNDB2* genes in response to salinity stress suggests that, despite the inherently high activities of AOX and external NADH oxidation observed in chickpea leaf mitochondria (and some other legumes), a stress-responsive aspect of the alternative pathway is still beneficial to the plant, at least in some tissues or cells. A similar correlation was found from publicly available data for *L. japonicus*, an important model legume. Overall, this suggests that different AP components play different roles and potentially in different cell types in chickpea and other legumes. Correlations in the stress-responsiveness of *CaAOX1* and *CaNDB2* occurred despite the distant localization of these genes in the genome. This suggests that their individual promoters contain common regulatory elements that are relevant to the salinity response, and indeed these two genes do contain some previously reported regulatory elements seen in the promoters of AP genes of *G. max* and *A. thaliana* [45,46]. In contrast, the *CaAOX2* genes, *CaNDA3* and *CaNDB3*, which are located very close to each other showed very different transcriptional patterns. Following gene duplication, altered transcriptional control may have arisen via new regulatory elements.

Correlations between transcriptional up-regulation of *CaAOX1* and *CaNDB2* with photosynthesis rates during salinity stress imply that one role of these stress-responsive AP genes is in prevention of photoinhibition or protection/maintenance of photosynthesis, as seen in tobacco and Arabidopsis [18,19,56,57,58]. Interestingly, *CaNDC1* also showed a positive correlation with photosynthesis, although not statistically significant. Since the Arabidopsis NDC1 ortholog affects the redox state of plastoquinone pools in plastoglobules [59], it is possible that upregulation in salt-treated chickpea may assist electron transport in photosynthesis. Other mechanisms are also important for maintaining photosynthesis in salt-exposed chickpea. For example, the ability to exclude Na from photosynthetic cell types enabled salt-tolerant Genesis836 to maintain higher rates of photosynthesis than salt-sensitive Rupali [60]. Analysis of AP gene expression not only in individual tissues, but also in specific cell types and both in the presence and absence of stress, would greatly facilitate our understanding of this pathway.

## 4. Conclusions

In this study we have identified all components of the alternative respiratory electron transport chain in chickpea and determined their combined activities in mitochondria purified from shoots and roots. The results point to a far greater respiratory capacity of leaves compared to roots. We have also determined the effect of salt stress on the expression of AP genes and shown that *CaAOX1* and *CaNDB2* expression is coordinated under these conditions, as seen in some other species. The effect of this on AP activity in intact chickpea plants requires further investigation using mass spectrometry, especially given the inherently high rates of external NADH oxidation and constitutive AOX activity in isolated mitochondria.

To further explore potential roles of the expanded alternative pathway protein family in legumes, it would be useful to determine protein abundance of the different type II NAD(P)H dehydrogenases. This will require the production of isoform-specific antibodies to the various NDB and NDA proteins. It will also be important to determine more precisely the sub-cellular location, tissue and cell-specific patterns of expression of the multiple AOX isoforms and ND proteins in chickpea and other legumes, and how they impact plant growth and performance.

## 5. Materials and Methods

### 5.1. Plant Materials and Growth Conditions

For mitochondrial isolations, *C. arietinum* (cv. Hattrick) plants were grown in seedling trays for 2 weeks in a 1:1 sand:vermiculite mix and watered as needed with a standard nutrient solution (1 mM MES, 1.5 mM Ca(NO_3_)_2_, 0.5 mM MgSO_4_, 1 mM KH_2_PO_4_, 50 µM KCl, 25 µM H_3_BO_3_, 2 µM MnSO_4_, 2 µM ZnSO_4_, 0.5 µM CuSO_4_, 0.5 µM Na_2_MoO_4_, 1 µM NiSO_4_, pH 6.5), within a controlled-temperature greenhouse. On average, light levels were 300–500 µmol m^−2^ s^−1^ (PAR), with 14-h day-lengths and temperatures ranging between 26 and 32 °C during the day and 18 and 22 °C overnight. Chickpea accessions and growth conditions for the salinity experiment were as described previously [37].

### 5.2. Measurement of Type II NAD(P)H Dehydrogenase Activities in Purified Mitochondria

Mitochondria were purified from leaf and root tissues using a previously described method [25,61]. Leaf tissue (10–15 g) was homogenised using a Polytron (Kinematica, Switzerland), while root tissue (up to 80 g) was homogenised in batches using a mortar and pestle and acid-washed sand. The remainder of the protocol was as reported previously. Mitochondrial integrity was measured using an assay for cytochrome C oxidase, by monitoring the oxidation of reduced cytochrome C on a spectrophotometer [62] in the presence and absence of the detergent Triton X-100 (0.01%). Purified mitochondria were assayed for NAD(P)H oxidation using a Clarke-type oxygen electrode system coupled to O_2_View software (Hansatech Oxygraph). Mitochondria were suspended within the chambers in a standard reaction medium (0.3 M sucrose, 10 mM TES, 10 mM KH_2_PO_4_ and 2 mM MgCl_2_, pH 7.2) unless stated otherwise. Assays were conducted as previously described [63]. Briefly, external NAD(P)H oxidation assays were supplemented with ADP (1 mM) and either NADH or NADPH (1 mM) and initially conducted in the presence of calcium chelator EGTA (0.25 mM), before activating with the addition of CaCl_2_ (1 mM). To engage matrix located NADH dehydrogenases (i.e., Complex I and the NDA enzymes), internal NADH was generated via oxidation of the NAD-linked substrates malate and glycine, which are likely to be the predominant mitochondrial substrates in leaves [1]. Mitochondria were provided with either with a combination of malate (10 mM) and glutamate (10 mM) at pH 6.8 (conditions that prevent accumulation of oxaloacetate) or glycine (10 mM) at pH 7.2. All malate and glycine assays contained NAD (1 mM) and Complex I activity was subsequently inhibited by rotenone (25 µM) to estimate the combined rate of the internal NDA enzymes. Each biological replicate was measured using a single technical replicate, although assays were repeated if the results appeared unusual.

### 5.3. Identification of ND Genes in Chickpea and Other Legumes

To identify candidate genes encoding CaND proteins, sequences were extracted from the *C. arietinum (*CDC Frontier) Annotation Release 101 (NCBI; [64]), using BLASTp searches with ND protein sequences previously characterized from *A. thaliana* [4]. Where possible, genes were named according to the level of amino acid identity to *A. thaliana* proteins. Details of gene loci were collected from NCBI and used to determine proximity of all ND genes to each other. Protein sequences were used to identify potential target peptides using online predictive tools iPSORT [65], TargetP [66], ChloroP [67], MitoProtII [68] and Predotar [69].

*CaND* mRNA sequences were used to query the chickpea transcriptome database CTDB [70] and the Legume Information System LIS [71] for RNA contigs with high homology. Where potential splice variants were encountered, sequences for each variant were used in BLASTn searches within the RNAseq data to determine which was most likely expressed, but primers were also designed to test for the presence of each variant using RT-PCR. The RNAseq libraries were also used to examine tissue-specific and stress-response patterns of expression of all available *CaND* genes. Primers unique to each *CaND* gene were also designed for qRT-PCR analysis.

To identify potential orthologues of ND genes in other legumes, initially the non-redundant protein database from NCBI was queried using chickpea ND mRNA sequences, then resultant RefSeq mRNA sequences were used to query specific legume databases, to confirm gene localization. Databases included: Lotus Base ([72]; https://lotus.au.dk), *L. japonicas* Gene Expression Atlas ([73]; ljgea.noble.org/v2/), *M. truncatula* Gene Expression Atlas [74]; mtgea.noble.org/v3/), Legume Information System ([71]; https://legumeinfo.org) Legume IP ([75]; plantgrn.noble.org/LegumeIP/), Chickpea Transcriptome Database ([35]; www.nipgr.res.in/ctdb.html), SoyBase ([76]; soybase.org/soyseq) and PeanutBase ([77]; peanutbase.org).

### 5.4. Quantitative Real-Time Reverse-Transcriptase PCR

Four plants from each accession and treatment group (four biological replicates) were randomly selected. On days 5, 9 and 15 (T5, T9 and T15) after the final salt application, the two youngest fully developed leaves were snap-frozen in liquid nitrogen, pulverized into frozen powder then used for RNA extractions using a modified Trizol-like solution [37]. Following DNase treatment (NEB Biolab, England), 2 µg of RNA was reverse transcribed by Protoscript-II Reverse Transcriptase kit (NEB Biolab, England). Samples of cDNA diluted with water (1:10) were used for qRT-PCR analyses,

For qRT-PCR expression analysis, KAPA SYBR Fast Universal Mix (KAPA Biosystems, USA), was used in a Real-Time qPCR system CFX96 (BioRad, USA) according to a previously described protocol [25]. Primers were designed to unique regions of *C. arietinum* ND genes and where possible flanked predicted intron splice sites (Appendix A). Assays were conducted in duplicate. Expression levels of target genes were normalized relative to the geometric mean of two reference gene transcript levels: *Hsp90*, Heat shock protein 90 (GR406804) and *Ef1α*, Elongation factor 1-alpha (AJ004960) [78].

### 5.5. Salinity Experiment

The experiment was carried out as described previously [37]. Briefly, seeds were germinated in Petri dishes for five days and transplanted to pots (four seedlings per pot) with artificial inoculation of rhizobium (NodulAid, BASF, Australia). Plants were grown in a controlled-temperature greenhouse with 25 °C/20 °C day/night temperature and 16h LED Grow Lights (~PAR 500 µmol m^−2^ s^−1^) (Heliospectra AB, Sweden). Pots were watered to a consistent soil moisture level of 80% field capacity. For salt stress, a final concentration of 90 mM NaCl (based on available soil moisture at 80% field capacity) was applied in four increments over two days, then maintained until the end of the experiment. In control pots, an equivalent volume of water was used under the same schedule.

### 5.6. Gas Exchange Measurements

Estimations of leaf photosynthesis and dark respiration were made using a Li-Cor 6400 Portable Photosynthesis System equipped with the standard 6 cm^2^ leaf chamber and an RGB light source. Adjustable parameters were set to 400 ppm CO_2_ (incoming/reference line) at a flow rate of 200 µmol s^−1^ and a block temperature of 25 °C. IRGAs were matched each time a new leaf was clamped. Measurements of photosynthesis were conducted using the youngest fully developed leaf of the main stem (typically the 4th leaf from the apex), after equilibrating for several minutes at 500 µmol m^−2^ s^−1^. Before leaf-clamping, the plant was exposed to 500 µmol m^−2^ s^−1^ (PAR) in the glasshouse for 30 min using supplemental LED lighting (Heliospectra LX602C, Conviron, CA). Two leaflets were trimmed from the base of the leaf, to ensure that the clamp could seal firmly around the leaf stem. After photosynthesis (A) measurement, the chamber light was turned off for 6–10 min until A and stomatal conductance were steady, then a measurement of respiration in the dark was collected. Measurements were conducted on randomised plants, between 10 am and 4 pm. Three successive measurements were logged per leaf.

### 5.7. Statistical Analyses

IBM^®^ SPSS v25 was used for statistical tests and data plotted using Microsoft^®^ Excel. For comparison of root and leaf mitochondrial activities (n = 3), an independent samples Kruskal–Wallis analysis was used, with post-hoc Dunn–Bonferroni tests. For comparison of root and leaf transcript levels, Mann–Whitney U tests were applied to each gene (n = 5). To test the effect of salt treatment on gene expression levels (n = 4) and gas exchange measurements (n = 3), Mann–Whitney U tests were used except in the case of Figure 6, where a Kruskal–Wallis test was first applied to each gene at T9, to determine whether there was a significant transcriptional effect of salt, before pairwise comparisons for each cultivar were made using post-hoc Dunn–Bonferroni tests. For correlation analyses, bivariate correlations using Spearman rank tests were used.

## Figures and Tables

**Figure 1 ijms-21-03844-f001:**
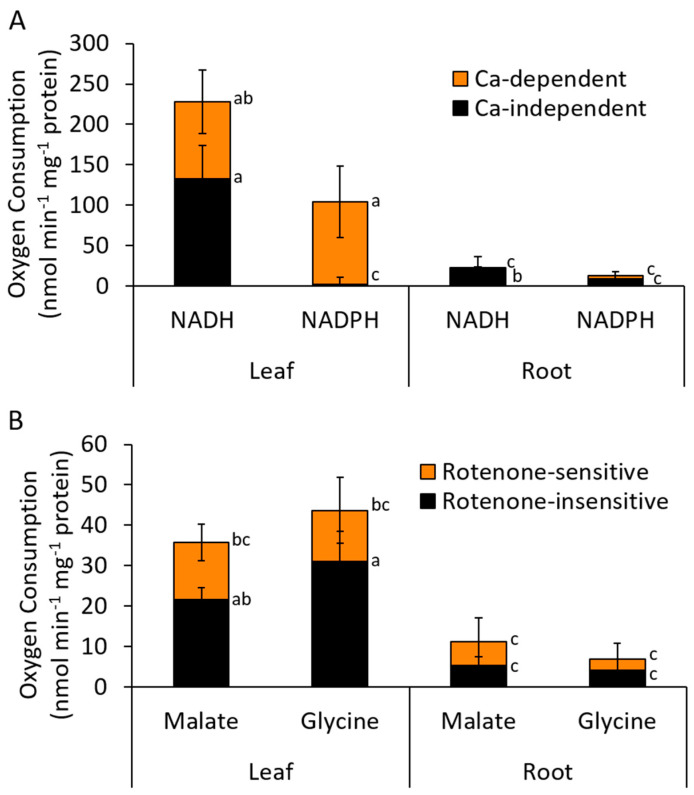
Activities of NAD(P)H dehydrogenases in chickpea leaf and root mitochondria. (**A**) External-facing NADH and NADPH dehydrogenase activities and their activation by calcium. (**B**) Internal-facing NADH dehydrogenase activities and their sensitivity to rotenone. Internal NADH was generated via the oxidation of the substrates, malate or glycine. Statistically significant differences indicated by differing letters, based on Kruskal–Wallis tests with post-hoc Dunn–Bonferroni tests (*p* < 0.05) (*n* = 3 ± S.E.M.).

**Figure 2 ijms-21-03844-f002:**
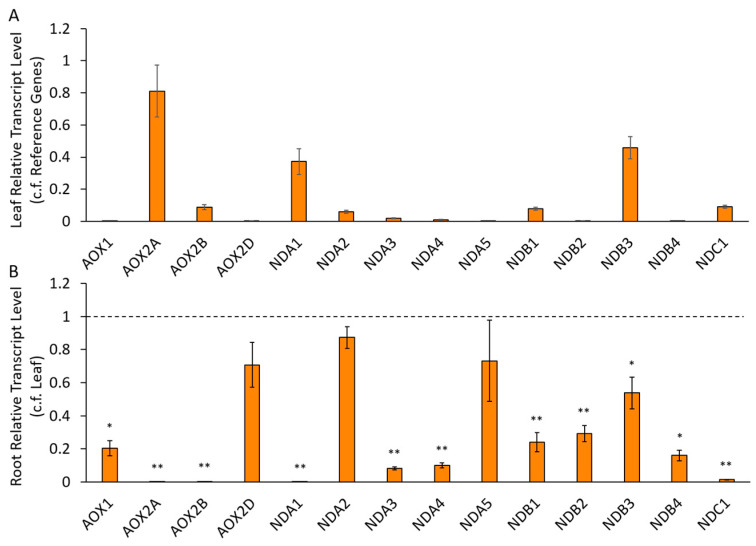
Expression of chickpea alternative pathway (AP) genes in leaf and root samples, using qRT-PCR. Transcripts are expressed (**A**) in leaf samples relative to transcript levels of two reference genes and (**B**) for root samples as a proportion of leaf samples. Statistical significance indicated by * (*p* < 0.05) and ** (*p* < 0.005) based on Mann–Whitney U tests between leaf and root samples, for each gene. (*n* = 5 ± S.E.M.).

**Figure 3 ijms-21-03844-f003:**
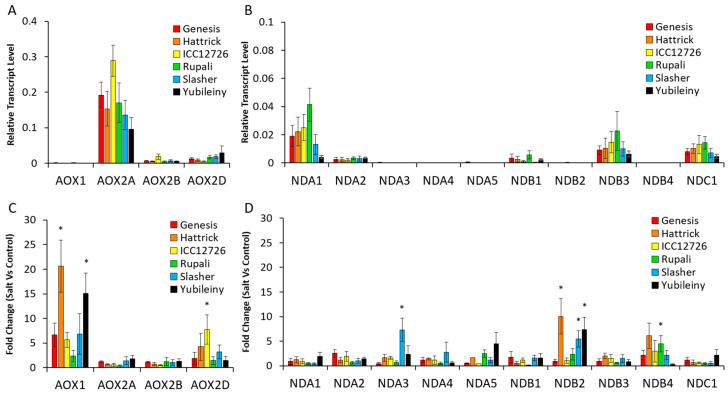
Expression of AP genes in control conditions and in response to salt treatment. Transcriptional patterns were expressed relative to reference genes for control samples in order to look for expression level differences between genes (**A**,**B**) and as fold change of salt-treated compared to untreated samples so that all genes could be visible on the same axis (**C**,**D**). Data were plotted from a single timepoint for clarity, at 9 days after salt treatment. This captured the peak response for most genes and cultivars. Data from 5 and 15 days after salt treatment showed similar patterns between control samples but some varying responses in salt-treatment (Appendix A). Statistical significance indicated by * (*p* < 0.05) based on Kruskal–Wallis tests between control and salt-treated samples, with post-hoc Dunn–Bonferroni tests for individual cultivars. (*n* = 4 ± S.E.M.).

**Figure 4 ijms-21-03844-f004:**
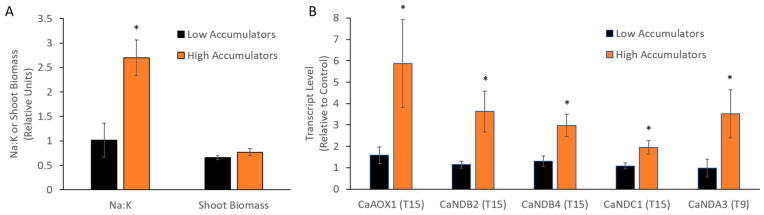
Transcriptional response of AP genes in cultivars with high or low leaf Na accumulation. Cultivars were grouped into low Na accumulators (Genesis836, Rupali and Yubileiny) and high Na accumulators (Hattrick, ICC12726 and Slasher), and compared for (**A**) Na:K and shoot biomass (n = 8) after 1 month of salt treatment and (**B**) fold change of AP genes relative to control plants (n = 4) at 15 days after salt treatment. Statistical significance indicated by * (*p* < 0.05) based on Mann–Whitney U tests between low and high Na accumulators.

**Figure 5 ijms-21-03844-f005:**
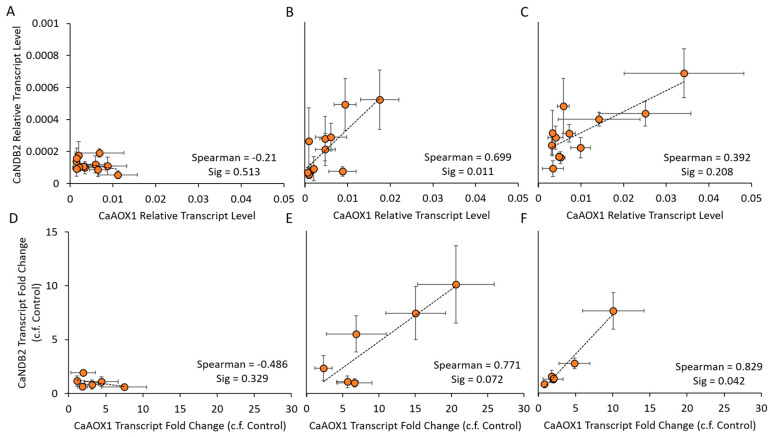
Co-ordinated expression of AOX1 and NDB2 in response to salinity stress in six chickpea cultivars over three timepoints. Comparisons were made based on relative transcript level data (**A**–**C**) and as a fold change of salt-treated plants relative to controls (**D**–**F**) at three different timepoints T5 (**A**,**D**), T9 (**B**,**E**) and T15 (**C**,**F**), using a Spearman correlation analysis in SPSS (v25, IBM). Means (n = 4 ± S.E.M.) plotted using Excel (Microsoft).

**Figure 6 ijms-21-03844-f006:**
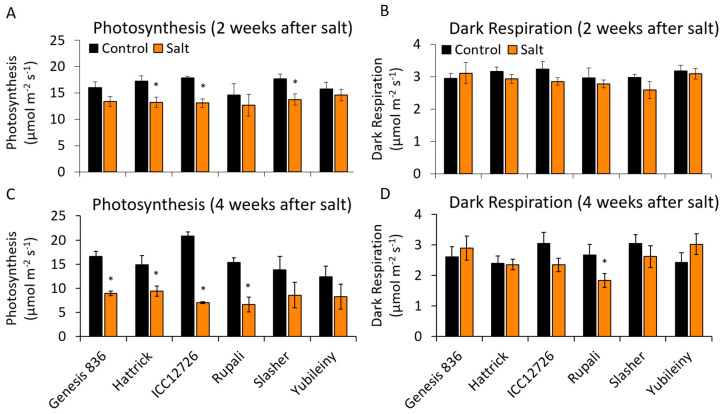
Effect of 2-week (**A,B**)and 4-week (**C,D**) salt exposure on leaf gas exchange parameters. Photosynthesis (**A,C**) was measured at 500 PAR and respiration (**B,D**) after 10 min equilibration in the dark. All measures were taken with 400 ppm CO_2_ in the reference line, block temperature of 25 °C and flow rate of 200 µmol s^−1^, using a LiCor 6400. Statistical significance indicated by * (*p* < 0.05) based on Mann–Whitney U tests between control and salt-treated plants. (n = 3 ± S.E.M.).

**Table 1 ijms-21-03844-t001:** Summary of chickpea type 2 NAD(P)H dehydrogenase (ND) gene nomenclature, accessions, gene locations and identities to the Arabidopsis orthologues.

Gene Name	Chromosome	Genome Location	Locus Tag	mRNA Accession	Arabidopsis Ortholog (% Identity)
*CaNDA1*	Ca5	25473001..25476933 complement	LOC101503077	XM_004500217	AtNDA1 and 2 (72.9 and 73.7)
*CaNDA2* *[CaNDA5]*	Ca2	25573447..25583926 complement	LOC101505730	XM_004490552 [XM_004490553]	AtNDA1 and 2 (72.4 and 73.0) [AtNDA1 and 2 (70.5)]
*CaNDA3*	Ca6	16203497..16207730	LOC101510826	XM_004504931	AtNDA1 and 2 (65.7 and 66.1)
*CaNDA4*	Ca4	31223913..31228505	LOC101502969	XM_004497885	AtNDA1 and 2 (65.9 and 65.8)
*CaNDB1*	Ca6	55894321..55900219	LOC101508266	XM_004507164	AtNDB1 (69.5)
*CaNDB2*	Ca1	1865091..1871364 complement	LOC101508897	XM_012713291	AtNDB2 and 3 (71.6 and 70.6)
*CaNDB3*	Ca6	15947443..15953377 complement	LOC101502575	XM_012717034	AtNDB2 and 3 (71.3 and 71.0)
*CaNDB4*	Ca6	55911093..55914789	LOC101508589	XM_004507165	AtNDB2-4 (66.3–67.5)
*CaNDC1*	Ca6	6799526..6805435 complement	LOC101503757	XM_004503838	AtNDC1 (67.8)

**Table 2 ijms-21-03844-t002:** *CaND* expression in shoot (or leaf) vs. root tissues. Data presented as raw FPKM from RNAseq transcriptomic datasets. Full datasets for other tissues can be seen in Appendix A. Veg = vegetative stage, Rep = reproductive stage, Sen = senescence stage.

Gene Name	Garg et al. [35], FPKM		Kudapa et al. [36], FPKM					
	15 d.o. Shoot	15 d.o. Root	Veg. Leaf	Veg. Root	Rep. Leaf	Rep. Root	Sen. Leaf	Sen. Root
*CaNDA1*	81	0	54	0.4	69	0.3	73	0
*CaNDA2*	13	24	16	39	17	26	22	25
*CaNDA3*	-	-	-	-	-	-	-	-
*CaNDA4*	0	0	14	2	8	6	9	2
*CaNDB1*	63	0	20	9	38	17	49	26
*CaNDB2*	-	-	-	-	-	-	-	-
*CaNDB3*	125	260	77	105	56	59	65	47
*CaNDB4*	-	-	-	-	-	-	-	-
*CaNDC1*	47	0	71	5	49	6	45	5

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
