# Peer review of "Identification of Alternative Mitochondrial Electron Transport Pathway Components in Chickpea Indicates a Differential Response to Salinity Stress between Cultivars"

_ijms, 2020, doi:10.3390/ijms21113844_

Round 1

Reviewer 1 Report

This is a well-written manuscript. I really enjoy reading this manuscript. Introduction and Abstract clearly identify the relevance of this research. As far as my experience goes, the methodology targets the main aim appropriately. The results are logically presented and are justified by the data which is presented in three tables and six figures. The discussion is easy to follow.

Author Response

Dear Reviewer 1,

Thank you for your review of our manuscript. Although you did not request changes to the original manuscript, we have made several improvements and submit a revised version.

Reviewer 2 Report

Manuscript title:

Identification of alternative mitochondrial electron transport pathway components in chickpea and their response to salinity stress in six different cultivars.

Authors:

Sweetman, C. et al.

The authors prepared mitochondrial fractions from the leaf and root tissue of chickpea (Cicer arientinum L.) to analyze enzymatic properties of type II NAD(P)H dehydrogenase (ND) as well as expression profiles of those enzyme genes in connection with those of alternative oxidases (AOX). They found that activities of both external and internal ND on inner mitochondrial membrane were higher in leaves than in root (non-photosynthetic tissue), and those tendencies were reproduced at the transcriptional levels of the genes as witnessed by their own experimental data and RNAseq of data in the cited references. Using six cultivars of chickpea that showed different tolerance to salt treatment, they showed that AOX1 and NDB2 were upregulated in response to salinity stress, but little evidence could be obtained for relationship between alternative pathway (AP) gene response and plant biomass under the salt treatment.

General comments:Contents and scientific value of the manuscript reached a certain satisfaction level, but this reviewer think that the manuscript need to be extensively improved. Using the transgenic Arabidopsis which overexpressed NDB2 and AOX1A or lacked the both gene expression, the same authors very nicely showed the roles of AP genes under the stress conditions (drought and strong-light) and provided valuable discussion (Plant Physiol. 2019, 181:774-788), but in the current manuscript, focal point of discussion was obscure, therefore very difficult to follow.

On the basis of data from Fig. 4 and Fig. S5, the authors argued that there might be common regulatory element for CaNDB2, CANDB3 and CaAOX1 genes despite they located distantly each other, while CaOX2, CaNDA3 and CaNDB3 gene might have different regulatory elements but they were locating very closely (L. 328-333). However, values of X and Y axis in Fig. 4 and Fig. S6 (may be the data of line and row in Fig. S5) were mixture of data at three points (5, 9 and 15 days) after the salt treatment, meaning that those data were mixture of sample data from three different populations. Data from different chronological point cannot be mixed, even though they are time-series data after single treatment. Biological status of the plants on each day after salt treatment differ from each other. The authors should separately obtain the correlation coefficient from respective points. Furthermore, Pearson’s correlation analysis requires that data of both axes (X and Y axis) are normally distributed, otherwise there will be no guarantee for robustness of the analysis. Looking at the data of Fig. 4 and Fig. S6, the distribution pattern of values in both axes were biased toward the origin on the graph and might not be normally distributed. If authors could not verify their data were normally distributed, they should follow non-parametric test (eg. Spearman’s rank correlation test).   

This reviewer thinks that the authors should re-examine (recheck) all the statistical methods used in the current manuscript: number of biological replicates (n=3) was not enough to perform a parametric statistical test, in such case, non-parametric statistical methods are preferable. How about the number of technical replicates in each experiment?

Application of salt to the plants was carried out at the concentration of 90 mM. Why did the authors use this concentration? Or did they check other concentrations elsewhere? Since data of the response of chickpea plant against salt was not available at this moment (ref.# 37 is in press) and Table S3 only showed the summary of performance of each cultivar, more detail information was required to understand salt stress response of chickpea.

Specific comments:

L.12: In chickpea there…      Scientific name is required. In chickpea (Cicer arinetinum) there…

L.16: external (cytosol)-..      Incorrect, external (inter-membrane space)-…

L.68 and 69: min-1mg-1 and…  min-1mg-1 protein and….

L.84: Arabidopsis thaliana     A. thaliana (full scientific name already appeared in L.50.)

Since full spelling of genus name are needed for the first appearance, the authors should check it again for other plant materials in the manuscript.

Fig.1-B: bc (malate-leaf), bc (glycine-leaf), c (malate-root), c (glycine-root), letters on top of bar graph. Are they ab, ab, b, b?

L.109-116:                  This section needs a citation of "Table 1" somewhere.

Fig.2: Data in A showed relative expression levels against reference genes (EF1-α and HSP90), but those in B were ratios against data in A. This is very confusable, especially for comparing the data between A and B. Data in B should be set up the same manner as A. No need to adjust Y axis of A and B same scale.

L.197: and AP gene expression response after one month of salt (not shown).

Does this mean salt treatment to plants was continued for one month? There was description “final NaCl application” (L. 179). Please make sure this point.

L.252: (adjusted R2=..)  “2” should be superscript.

L369-: Although the methods for determination of enzymatic activities were reported previously, more detail descriptions were required; as role of alternative pathway and its metabolism is complicated process, a brief descriptions of reason why the authors used “malate” and “glycine” as substrates of the reactions for inner ND will help readers well understand this process.

Author Response

Dear Reviewer 2,

Thank you for your thorough review of our manuscript. We have addressed your feedback in the attached document.

Round 2

Reviewer 2 Report

Comments for revised manuscript of ijms-799463

Revised manuscript title:

Identification of alternative mitochondrial electron transport pathway components in chickpea and their indicates a differential response to salinity stress between in six different cultivars.

The revised manuscript was improved well and almost acceptable for publication in IJMS. However, this manuscript still has problems in descriptions about statistics.

Fig. 1: As number of biological replicates of this experiment was 3, the authors used Kruskal-Wallis test (nonparametric ANOVA) for multiple comparison of the means. This is OK. However, there was no description about the method for pairwise comparison test among respective means. Since they used IBM SPSS, this reviewer supposed that Bonferroni correction was applied to the test. Please specify the name of multiple comparison test (Dunn test? Dunn-Bonferroni test? Steel-Dwass test?).

Figs. 2, 3, 4, 6, Fig. S3: As the authors described in their previous manuscript that unpaired t-test between two samples was carried out (legends of figures), it meant that comparison was made for between two sample (eg. leaf and root sample, control and salt-treated sample, etc.). However, in the current manuscript, Kruskal-Wallis test was used for the comparison. This reviewer thinks that the authors mistakingly used term “Kruskal-Wallis test”, instead of term “Mann-Whitney U test or Dunn test”. Please correct them.

  1. 116. “E-F hand” should be “EF-hand”
  2. 441. “Cicer arientinum” is “C. arientinum
